# A New Benchmark and Reverse Validation Method for Passage-level Hallucination Detection

**Shiping Yang**[*†] **, Renliang Sun**[*]**, Xiaojun Wan**

Wangxuan Institute of Computer Technology, Peking University

Center for Data Science, Peking University

The MOE Key Laboratory of Computational Linguistics, Peking University

yangshipingnlp@gmail.com

sunrenliangpku@gmail.com

wanxiaojun@pku.edu.cn

## Abstract

Large Language Models (LLMs) have shown their ability to collaborate effectively with humans in real-world scenarios. However, LLMs are apt to generate hallucinations, i.e., makeup incorrect text and unverified information, which can cause significant damage when deployed for mission-critical tasks. In this paper, we propose a self-check approach based on reverse validation to detect factual errors automatically in a zero-resource fashion. To facilitate future studies and assess different methods, we construct a hallucination detection benchmark named PHD, which is generated by ChatGPT and annotated by human annotators. Contrasting previous studies of zero-resource hallucination detection, our method and benchmark concentrate on passage-level detection instead of sentence-level. We empirically evaluate our method and existing zero-resource detection methods on two datasets. The experimental results demonstrate that the proposed method considerably outperforms the baselines while costing fewer tokens and less time. Furthermore, we manually analyze some hallucination cases that LLM failed to capture, revealing the shared limitation of zero-resource methods.

## 1 Introduction

Recent research has shown that Large Language Models (LLMs), like ChatGPT and GPT-4 (OpenAI, 2023), have achieved state-of-the-art across various NLG tasks (Bubeck et al., 2023; Kocoń et al., 2023; Qin et al., 2023). Despite their remarkable capabilities in various aspects, LLMs still suffer from some inherent drawbacks, underperforming in scenarios involving complex reasoning (Frieder et al., 2023; Chen et al., 2023) and global learning (Li et al., 2023a).

One prominent weakness of LLMs is their tendency to hallucinate when generating fluent and informative responses. This hallucination issue significantly undermines public trust in LLMs and limits their deployment in certain critical and sensitive domains such as legal (Huang et al., 2023), medical (Singhal et al., 2023), and finance (Wu et al., 2023).

Some recent studies have attempted to alleviate the hallucination of LLMs. They emphasized that knowledge gaps in LLMs are a primary cause of hallucination (Petroni et al., 2019; Lee et al., 2022; Zheng et al., 2023; McKenna et al., 2023), which has inspired a retrieval strategy to mitigate hallucination through interacting with external knowledge bases (Guo et al., 2022; Peng et al., 2023). Moreover, there is a growing research in zero-resource hallucination detection (Manakul et al., 2023; Cohen et al., 2023). However, in our view, previous studies suffer from the following two disadvantages: **(1) Relying on external retrieval modules.** Retrieving external knowledge often has complex processes and notable delays. In addition, external databases may also be inaccessible. **(2) Only focusing on sentence-level hallucination detection.** Real-world applications often require passage-level hallucination detection rather than sentence-level detection. This arises from the fact that LLMs tend to furnish users with comprehensive and informative answers instead of a single sentence.

In many scenarios, a judgment about the entire passage is enough, which enables a quick decision on whether to activate the retrieval module and generate a new response. It's highly inefficient and costly to access the truthfulness of a response sentence by sentence. Therefore, exploring passage-level hallucination detection holds greater practical significance than exclusively concentrating on sentence-level detection. Unfortunately, research on passage-level hallucination detection is still scarce: to the best of our knowledge, there is no available method and suitable benchmark currently.

---

[*] Equal contribution

[†] Work done during internship

To facilitate research on passage-level hallucination detection, we first annotate and propose the PHD (**P**assage-level **H**allucination **D**etection) benchmark, a new benchmark for the evaluation of passage-level detection methods. We then replicate some recent sentence-level detection methods and adapt them to passage-level detection. Due to the poor performance of these methods on our benchmark, we propose a zero-resource hallucination detection method named **Reverse Validation (RV)**. RV is specifically tailored for passage-level detection, which can detect whether a passage contains factual errors without access to token-level probability distribution and external knowledge.

We empirically evaluate the RV method on the PHD benchmark and the WikiBio-GPT3 dataset (Manakul et al., 2023). The results show that the RV method significantly outperforms existing methods and baselines, demonstrating competitive performance, lower token costs, and faster response times. In addition, we conduct an ablation study to explore the robustness and compatibility of our approach for different LLMs. Finally, we reveal the shared limitation of zero-resource detection methods by investigating bad cases.

We are committed to facilitating research on passage-level hallucination detection. Our contributions can be summarized as follows: **(1)** We create a high-quality benchmark named PHD for the evaluation of passage-level hallucination detection methods, and it will become a challenging benchmark for hallucination detection. **(2)** We propose the reverse validation method to detect passage-level hallucinations, which can be used in black-box models and zero-resource fashion. **(3)** We demonstrate the effectiveness and robustness of the proposed method by comparing it with existing methods on two datasets. Data and code will be released at https://github.com/maybenotime/PHD.

## 2 Related Work

### 2.1 Hallucinations of LLMs

Hallucination has been studied in different natural language generation tasks (Dziri et al., 2022; Gupta et al., 2021; Maynez et al., 2020; Raunak et al., 2021), and its definition may vary depending on the specific task. A recent survey categorized hallucinations into two classes (Ji et al., 2023), intrinsic hallucinations and extrinsic hallucinations. Intrinsic hallucinations refer to generated output that contradicts the original content, while extrinsic

hallucinations refer to output that cannot be confirmed or contradicted by the source. According to a recent study (Bang et al., 2023), intrinsic hallucinations are rarely observed in LLMs such as ChatGPT, whereas extrinsic hallucinations tend to occur more frequently. Therefore, when referring to hallucinations in LLMs, we primarily focus on extrinsic hallucinations.

It is worth noting that extrinsic hallucination doe not always involve errors, as it can originate from factually correct external information (Maynez et al., 2020; Thomson and Reiter, 2020). Such factual hallucination can be helpful because it recalls additional background knowledge from the parametric knowledge base of LLMs to generate informative responses. Hence, in recent work on LLMs, the term "Hallucination" is only used to describe the phenomenon where the model generates non-factual statements or fabricates unverifiable knowledge (Manakul et al., 2023; Peng et al., 2023; Li et al., 2023b; Cohen et al., 2023).

### 2.2 Mitigate Hallucinations

With the rise of ChatGPT, there has been an increased awareness of its tendency to generate plausible-looking statements that are unverifiable or factually incorrect. To alleviate this issue, there are many active endeavors to analyze the causes of hallucinations and mitigate the hallucination of LLMs.

Zheng et al. (2023) hypothesized that the hallucinations may be attributed to knowledge gaps in LLMs, and several works have shown the promise of using retrieval knowledge to mitigate them (Lewis et al., 2020; Shuster et al., 2021; Peng et al., 2023). A natural idea is whether LLMs can detect hallucinations in the outputs without any external knowledge, only invoking the retrieval module when hallucinations occur to shorten the response time. This zero-resource method can also serve as an alternative solution in situations where external databases are inaccessible.

One approach to detect hallucination without external knowledge is by utilizing intrinsic uncertainty metrics, such as token probability and perplexity. However, the token-level probability distribution is inaccessible for current LLMs like ChatGPT.

Therefore, there is a growing interest in applying zero-resource hallucination detection for black-box models (Manakul et al., 2023; Cohen et al., 2023).

Our work is closely related to these works, but we focus on passage-level zero-resource hallucination detection instead of sentence-level.

## 3 The PHD Benchmark

In this section, we describe the detailed information of our PHD Benchmark. Specifically, we show how we select entities to generate hallucination data in Section 3.1. Then, we give the annotation process of PHD benchmark in Section 3.2. Finally, we report statistical information about our PHD benchmark in Section 3.3.

### 3.1 Data Generation

A primary use case of LLMs is answering questions and seeking factual knowledge about an entity. Wikipedia contains plenty of entities and corresponding introduction articles, which serve as a commonly used data source for constructing fact verification datasets (Thorne et al., 2018). Hence, we extract entities from the Wikipedia dump[1] to generate passages and employ the associated Wikipedia articles as references at the following annotation round.

We choose ChatGPT, currently the most widely used and representative LLM, to generate our hallucination data. However, as one of the most powerful LLMs, ChatGPT rarely generates hallucinations when responding to common questions, resulting in an insufficient number of hallucination samples included in the benchmark. Previous work has addressed this issue by instructing LLMs to generate hallucinations intentionally (Li et al., 2023b) or to generate data on specific topics to which the model is prone to hallucinate (Manakul et al., 2023). However, a benchmark that can effectively evaluate the performance of hallucination detection methods should contain entities from various domains and hallucinations generated by models naturally, which better match the hallucination scenarios methods may encounter in practical usage. Therefore, we propose a different strategy to generate as many hallucinated samples as possible in the benchmark.

A factual error often occurs when a model lacks sufficient knowledge (Zheng et al., 2023). This phenomenon inspires us to generate hallucinations by selecting entities that lack enough facts in the

model. By accessing the data volume of a specific entity within the training data, we can readily find entities that LM might not be acquainted with. Although ChatGPT's training data is not accessible, publicly available information is that it includes the C4 corpus (Raffel et al., 2020) and a large amount of data crawled online. Consequently, we propose a simple method to proxy the data volume of an entity in the training data through the number of related items that Google Search returns. Then, we create a dataset consisting of entities and their corresponding proxies of data volume. We categorize the entities into three domains based on distinct data volumes: *PHD-Low* (<100K), *PHD-Medium* (100K~1M), and *PHD-High* (>1M), respectively.

We randomly sample 100 entities from each of the above domains to construct our PHD benchmark. Then, we prompt ChatGPT[2] to generate a Wikipedia article about each entity using the prompt *Please write a brief Wikipedia for {Entity}*. Following these procedures, we have successfully prepared the data for annotation in the next stage.

### 3.2 Human Annotation

Annotating the hallucination at the passage level is a very challenging task for human labelers, which requires workers to search for evidence to check each claim within the passage, often from a long document and various websites. Therefore, we design a two-stage human annotation process and select reliable workers through strict criteria to ensure the quality of PHD benchmark.

**Labels Definition** The definition of hallucinations has been introduced in detail in Section 2. Following the previous work, we define three labels used in our annotation process: (1) *factual*: every claim in the passage is supported by evidence. (2) *non-factual*: the passage consists of non-factual or unverifiable information (even a minor error). (3) *unverifiable*: a label only used in the first stage when there is insufficient evidence to reject or support claims in the passage by using Wikipedia as a reference.

**Worker Requirements** The requirements for workers who perform annotation are as follows: (1) had education experience in university with at least a bachelor's degree; (2) passed the TOEFL or IELTS exam; (3) good at using search engines

---

[1]The English Wikipedia dumps are from `https://archive.org/download/enwiki-20210820`. The version we used is 2021-08-20, which is released before the deadline for training data of ChatGPT.

[2]We set the temperature to 0.0 when generating data to reduce randomness.

to look up reliable information; (4) passed the corresponding Qualification Test (QT) designed for our task (more details below). These requirements are to ensure that workers are proficient in English reading comprehension and can perform the hallucination annotation task.

**Qualification Test** We designed a Qualification Test (QT) to measure the worker's hallucination detection ability. Before the QT, workers were required to read the task guide about how to annotate the hallucination. We provided a detailed explanation of the passage-level hallucination detection task. After workers took the QT, all submissions were manually checked to filter out workers who could not perform annotation correctly. Finally, ten candidates took the QT, and three of them passed the test and entered the following annotation stage.

**Two-stage Annotation** The annotation task was divided into two stages. In the first stage, qualified workers were asked to annotate hallucination using the corresponding Wikipedia articles as references. If workers fail to gather information from Wikipedia to support a claim, they will annotate the passage as unverifiable. In the second stage, we let workers re-annotate unverifiable passages through access to the Internet. The content they browsed was obtained from the official website on the first page of Google search results to ensure the reliability of the evidence. When the second stage finished, all the passages were classified as factual or non-factual. Except for annotating labels, workers were required to mark the corresponding content they believed to be incorrect or unverifiable. This additional information can provide help in the evaluation of fine-grained hallucination detection.

**Quality Control** We added some fake examples to ensure the quality of the benchmark. Concretely, some fake examples annotated by researchers were assigned to workers in each annotation stage. We use Cohen's Kappa (k) to measure their agreement with the labels of researchers and obtain an average k=0.848 (0.80<=k<=1.00), showing a perfect consistency. We also compute Fleiss' Kappa (Fleiss, 1971), which achieves 0.77.

### 3.3 Statistics

As we introduced above, the hallucination data was generated by ChatGPT and subsequently annotated by workers. Table 1 shows the final annotation result of the PHD benchmark.

Figure 1: The overview process of the reverse validation (RV) method

The entities of the *PHD-High* domain rarely contains hallucinations in their passages, while nearly half of the passages of entities in the *PHD-Low* domain hallucinate. The different distributions of hallucinated samples in three domains of PHD support the previous hypothesis that hallucinations are commonly caused by knowledge gaps in LLMs (Zheng et al., 2023; Ji et al., 2023).

|  | Factual | Non-factual |
|---|---|---|
| *PHD-High* | 86 | 14 |
| *PHD-Medium* | 76 | 24 |
| *PHD-Low* | 60 | 40 |

Table 1: The annotation result of PHD benchmark

## 4 Method

In this section, we propose a new method named Reverse Validation (RV). RV can be used to detect whether passages generated by LLMs contain hallucinations.

### 4.1 Reverse Validation Method

Our method is motivated by the insight of understanding language models as knowledge bases (Petroni et al., 2019). In brief, we regard LLMs as huge databases that store entities and related knowledge, capturing factual errors by construct-

ing a prompt to retrieve the corresponding entity from LLMs. We next describe the three stages of using this method, with the overall process illustrated in Figure 1.

**Stage 1: Construct Query** We extract the entity of user concern from the response and use the remaining information to construct a query.

**Stage 2: Access Databases** Then, We use the query we construct in the first stage to access "databases" and get a corresponding answer.

**Stage 3: Entity-Answer Match** Finally, we assess whether the returned answer matches the initial entity. If there is a match, it indicates that the response about the entity is factual. Otherwise, the response contains hallucinated information.

Stages 1 and 2 are processes of reverse probability modeling, which can be defined as follows: Given a user query $Q$, the process of generating responses can be formulated as $\mathbf{P}\left(\mathbf{R}|\mathbf{Q}\right)$, where $R$ denotes the response. This formula can be further simplified as $\mathbf{P}\left(\mathbf{I}|\mathbf{E}\right)$, where $E$ denotes the entity we extracted, and $I$ denotes the remaining information. The process of constructing a query to access LLMs can be denoted as $\mathbf{P}\left(\mathbf{E}|\mathbf{I}\right)$, which indicates generating the corresponding entity with hard constraints. After we complete the reverse modeling to obtain the corresponding entity, we validate it through entity-answer matching to reach a final decision. Therefore, we named our method **Reverse Validation (RV)**.

The RV method works based on the fact that if a response contains hallucinations, it becomes an incorrect search condition when we reconstruct it as a query to access LLMs. Wrong retrieval conditions will lead to LLM generating a wrong entity or even failing to find the relevant entity. In contrast, if the retrieval is successful, it indicates that the entity information is stored in parameterized knowledge rather than a product of hallucinations.

### 4.2 Implementation of Variants

Recent success in prompting enables us to query LLMs with human language (Brown et al., 2020; Liu et al., 2023). Therefore, our method can be implemented by designing suitable prompts. Specifically, We design two variants of our RV method, i.e., two different ways of constructing a query. The prompts used for each variant at every stage of the RV method are shown in Table 2.

**Reverse Validation via Question Generation** This variant constructs a query using question generation (QG). We prompt LLMs to generate a question about the entity based on the content of the response. One key point of implementation is that the entity name cannot appear in the question, which will lead to a label leakage.

**Reverse Validation via Entity Matching** This variant constructs a query by instructing the LLMs to perform the entity matching (EM) task directly. We rewrite the relevant information into a list of requirements and then request the LLM to return an entity that can match the requirements. Considering that the model may return an entity that meets part of the requirements when it fails to locate a perfectly matched entity, we require the model to report the percentage of requirements the entity satisfies while returning the entity. This percentage plays a crucial role in the entity-answer match stage, as only a match exceeding 90 percent can be recognized as factual. Furthermore, we observe that LLMs will point out the requirements that are not met by the returned entity, thereby enhancing the interpretability of our method.

For both variants, we use an exact string matching strategy to implement the third stage of the Reverse Validation method.

## 5 Experimental Setup

In this section, we introduce the model, the datasets, the evaluation metrics, and the methods that are compared in the experiments.

**Model** We use a stable **ChatGPT** version (gpt-3.5.turbo-0301[3]) and set the generation temperature to 0.0 in our experiments, which ensures the reproducibility of results to the greatest extent.

**Datasets** In addition to testing on the PHD benchmark, our evaluation also incorporates the synthetic WikiBio dataset (Manakul et al., 2023). This dataset comprises 238 passages generated by GPT3 and annotated at the sentence level, with WikiBio (Lebret et al., 2016) serving as the reference source. While the dataset lacks passage-level labels, we can aggregate sentence labels to derive pseudo-labels at the passage level. There are very few completely correct passages in this dataset. Consequently, we extract a subset of the synthetic dataset,

---

[3]https://platform.openai.com/docs/models/gpt-3-5

| Stage | Reverse Validation via QG | Reverse Validation via EM |
|---|---|---|
| (1) Construct Query | I will give you some information about the entity. You should use all this information to generate a question, and the answer to your question is the entity. Do not include the entity in your question. {*Example*} Entity: {*Entity*} Information: {*Response*} Question: **\<Question>** | {*Response*} Please list all features of {*Entity*} which are mentioned above with numbers, do not include {*Entity*} in your list. **\<Requirements>** |
| (2) Access Databases | You should answer the following question as short as possible. {**\<Question>**} | You should find an entity that conforms to the following description: {**\<Requirements>**}. If you fail to find a perfect match, please say an entity that matches the requirements as much as possible. You need to give the percentage of the entity that meets requirements.' |

Table 2: Prompts used for each variant at every stage of our **RV** method. **\<Question>** and **\<Requirements>** are generated by models in the first stage.

consisting of 200 passages that are determined to be non-factual. In our experiment, we referred to this subset as the WikiBio-GPT3 dataset.

**Evaluation Metrics** We evaluate how well the method detects passages that are non-factual using the following metrics:[4] **Precision**: the portion of non-factual passages out of the passages rejected by the detection method; **Recall**: the portion of passages rejected by the detection method out of all the non-factual passages; **F1**: the harmonic mean of precision and recall; **Accuracy**: the portion of passages rejected by the detection method out of all the passages. We only report Accuracy on the WikiBio-GPT3 dataset.

**Compared Methods** Considering that there is no method for passage-level hallucination detection, we replicate the existing zero-resource detection methods and modify them for passage-level detection. We also add two variations of our approach.

- **All False**: A straightforward approach that assigns the label "non-factual" to all the passages as a baseline.

- **LMvsLM revised**: LMvsLM, cited as the current leading method for detecting hallucinations at the sentence level (Cohen et al., 2023), is inspired by truth-seeking mechanisms in the legal field. It establishes a multi-turn interaction between an "examiner" LM and an "examinee" LM to capture contradictory statements. Following the setting in their experiments, we employed ChatGPT to fulfill different roles within the LMvsLM framework. However, when we applied this method directly to detect

hallucinations at the passage level, the "examiner" concluded that all passages were correct. Upon examining the interaction history between the two models, we discovered that the "examinee" LM answered questions based on the passage's content rather than retrieving relevant knowledge from its parameters. To resolve this issue, we revised the "examinee" LM's setup by modifying the original prompt "*Please answer the following questions regarding your claim: {Questions}*" to "*Please answer the following questions: {Questions}*" and restricted its access to the dialogue history.

- **SelfCheckGPT via BERTScore**[5]: A variant of SelfCheckGPT (Manakul et al., 2023), using BERTScore (Zhang et al., 2019) to measure the consistency between the response and stochastic samples. However, this method returns a sentence score between 0.0 to 1.0 instead of a label that indicates factual or non-factual. We derive passage-level scores by averaging the sentence-level scores. To align this method with our task, we then optimize a threshold over the WikiBio-GPT3 dataset. If a passage obtains a score higher than the threshold, it will be categorized as non-factual. We only report SelfCheckGPT via BERTScore on the PHD benchmark.

- **Zero-shot Detection**: Recent research has demonstrated the capability of LLMs to identify the presence of hallucinated information in various natural language generation tasks,

---

[4]The detection method will "rejects" a passage when identifying a passage containing hallucinations.

[5]We use the implementation of SelfCheckGPT released by the author at: https://github.com/potsawee/selfcheckgpt

even in a zero-shot manner (Li et al., 2023b; Cohen et al., 2023). We revised their instructions slightly to perform our passage-level detection task. We use the following prompt: *I want you to act as a claim judger. Given a claim about an entity, your objective is to determine if the provided claim contains non-factual or hallucinated information. You should give your judgment based on world knowledge, and answer with factual or non-factual. {Claim}.*

- **Reverse Validation via Question Generation (QG)**: A variant of our reverse validation method, using question generation to construct a query to access the parametric Knowledge base of LLMs.

- **Reverse Validation via Entity Matching (EM)**: A variant of our reverse validation method, instructing LLMs to perform the entity matching task to construct a query to access the parametric Knowledge base of LLMs.

Please refer to Section 4.2 for the concrete implementation of two variants.

## 6 Results

### 6.1 Main Results

Table 3 shows the result of our experiments on the PHD benchmark.

After revising the setting of **LMvsLM** to make it suitable for passage-level hallucination detection, this method achieve the highest Precision score among all the baselines. Nevertheless, it receives a lower score in terms of Recall.

In order to test **SelfCheckGPT via BERTScore** on the PHD benchmark, we optimize a classification threshold for hallucination detection using WikiBio-GPT3 dataset. However, the domain shift issue caused by different sources of two datasets (generated by different LLMs) makes the optimal threshold difficult to generalize to the PHD benchmark, which result in the poor performance of **SelfCheckGPT via BERTScore**. Interestingly, we observe that **Zero-shot Detection** completely fails to detect hallucination at passage-level, categorizing all the passages as factual labels.

Across all the domains of PHD, two variants of our method outperform the baselines, often by a large margin. Notably, the performance of **Reverse**

**Validation via EM** is better than **Reverse Validation via QG**, indicating that the percentage of requirements fulfilled by an entity can be helpful to make a better decision in the **Entity-Answer Match** stage.

While both variants of our method exhibit good performance in the *PHD-LOW* and *PHD-Medium* domains, they all struggle when it comes to the *PHD-High* domain. This phenomenon suggests that there might be an implicit relationship between the difficulty of hallucination detection and the data volume of entities in the training data.

In Table 4, we present the accuracy of all methods when evaluated on the WikiBio-GPT3 dataset. To measure whether our method is sensitive to fine-grained hallucinations, we also report the minimum value of the hallucination ratio[6] among the passages that have been rejected. Using this metric, we find that the two variants of our reverse validation method can detect fine-grained hallucinations, even when the passages only contain one non-factual sentence. In contrast, **LMvsLM revised** fail to identify a non-factual passage with a hallucination ratio below 60 percent.

Last, we calculate the average token cost and time delay for each method on the PHD benchmark. The results shown in Table 5 demonstrate that our methods have significant advantages in terms of token cost and response latency, which can be applied to production-level systems.

### 6.2 Ablation Study Results

We perform ablation study to understand the importance of the backbone models for RV method. Therefore, we employ **Llama-2-7b-chat-hf**, an open-source LLM, as the backbone model and conduct experiments [7] on the PHD benchmark. Results are reported in Table 6.

It is worth noting that the ablation experiments using **Llama-2-7b-chat-hf** are not conducted in self-check scenarios since the PHD benchmark was generated by ChatGPT. This change in the experimental setup considerably impairs the performance of **LMvsLM revised** and **SelfCheckGPT via BERTScore**.

For **LMvsLM revised**, the "examinee" Llama2 might provide answers that contradict the claim

---

[6]The hallucination ratio can be calculated by dividing the count of non-factual sentences in the passage by the total number of sentences in the passage. We calculate it for each passage in the WikiBio-GPT3 dataset.

[7]We set the temperature to 0.1, top_p to 0.05, top_k to 1.

|  | PHD | | | PHD-LOW | | | PHD-Medium | | | PHD-High | | |
|---|---|---|---|---|---|---|---|---|---|---|---|---|
|  | F1 | P | R | F1 | P | R | F1 | P | R | F1 | P | R |
| All False | 41.3 | 26.0 | 100 | 57.1 | 40.0 | 100 | 38.7 | 24.0 | 100 | 24.6 | 14.0 | 100 |
| LMvsLM revised | 25.5 | **75.0** | 15.4 | 25.5 | **85.7** | 15.0 | 34.5 | **100** | 20.8 | 11.1 | 25.0 | 7.1 |
| SelfCheckGPT via BERTScore | 41.1 | 26.0 | **97.4** | 56.1 | 39.4 | **97.5** | 39.7 | 24.7 | **100** | 23.6 | 13.5 | **92.8** |
| Zero-shot Detection | 0.0 | 0.0 | 0.0 | 0.0 | 0.0 | 0.0 | 0.0 | 0.0 | 0.0 | 0.0 | 0.0 | 0.0 |
| Reverse Validation via QG | 52.3 | 40.7 | 73.1 | 58.6 | 44.7 | 85.0 | 53.3 | 44.4 | 66.6 | 33.3 | 25.5 | 50.0 |
| Reverse Validation via EM | **59.1** | 48.0 | 76.9 | **63.6** | 50.7 | 85.0 | **64.2** | 58.6 | 70.8 | **41.9** | **31.0** | 64.3 |

Table 3: F1 scores, Precision (P), and Recall (R) for baselines and two variants of our **RV** method on the PHD benchmark. We also report the metrics on three subdomains of PHD benchmark. We use **Bold** to mark the best result and underline the second-best result.

generated by ChatGPT due to knowledge gaps between the two models. Consequently, the "examiner" classifies most claims as non-factual.

As a sampling-based approach, **SelfCheckGPT via BERTScore** using BERTScore to measure the consistency between the response and stochastic samples. When a different model is employed for sampling, the generated samples can diverge significantly from the original response, resulting in this method devolving into an "All false" baseline. Therefore, this method is only suitable for self-check scenarios.

Both **ChatGPT** and **Llama-2-7b-chat-hf** fail to detect passage-level hallucinations in a zero-shot setting. This phenomenon demonstrates that passage-level detection is more challenging than sentence-level detection for LLMs, particularly when the hallucination is generated by models themselves.

Unlike ChatGPT, **Llama-2-7b-chat-hf** is a smaller and less powerful LLM, and one would expect degraded performance with **Llama-2-7b-chat-hf** as the backbone model. The results of **RV via EM** align with our expectations. However, we observe that the performance of **RV via QG** remains relatively stable when the backbone model is changed, which indicates the robustness of this variant.

Contrasting with other baselines, our methods are effective not only in self-checking scenarios but also in situations using other LLMs. We can even achieve competitive performance using a small LLM like **Llama-2-7b-chat-hf**. In summary, our reverse validation method shows strong robustness and compatibility across different settings.

## 7   Case Study and Analysis

We manually analyzed the false negative and false positive examples of our method to give a better

|  | Accuracy | Min Hal. Ratio |
|---|---|---|
| LMvsLM revised | 29.8% | 60.0% |
| Zero-shot Detect. | 0% | N/A |
| RV via QG | 74.2% | 10.0% |
| RV via EM | 67.2% | 10.0% |

Table 4: The result of our experiments on the WikiBio-GPT3 dataset. *Min Hal. Ratio* means the minimum **Hallucination Ratio** among the passages that have been rejected.

|  | Token Cost | Time Delay (sec.) |
|---|---|---|
| RV via QG | 297.3 | 2.65 |
| RV via EM | 510.4 | 7.25 |
| LMvsLM rev. | 5974.5 | 29.48 |
| SelfCheckGPT. | 435.7 | 122.4 |

Table 5: Average token cost and time delay of different methods on the PHD benchmark. We use SelfCheckGPT. to represent **SelfCheckGPT via BERTScore**.

understanding of the failure cases.

### 7.1   False Negative

Table 7 provides two instances of false negatives, which were identified as hallucinations mistakenly by **Reverse Validation via EM**.

For the entity "Maximilian II Emanuel", LLM answers an abbreviation rather than its full name. In another case, the model provides a general entity "Jordan" instead of a more specific entity "History of Jordan". These situations are very common in our method. However, our automatic matching process, which relies on exact string matching, categorizes all these cases as non-factual due to a minor string discrepancy.

The poor performance of exact string matching is the primary factor causing false negatives. In order to address this issue, we employ LLM to perform fuzzy matching, using the prompt *Please*

| | PHD | | | PHD-LOW | | | PHD-Medium | | | PHD-High | | |
|---|---|---|---|---|---|---|---|---|---|---|---|---|
| | F1 | P | R | F1 | P | R | F1 | P | R | F1 | P | R |
| All False | 41.3 | 26.0 | 100 | 57.1 | 40.0 | 100 | 38.7 | 24.0 | 100 | 24.6 | 14.0 | 100 |
| LMvsLM revised | 41.5 | 26.3 | **98.7** | 57.6 | 40.4 | **100** | 40.0 | 25.0 | **100** | 23.2 | 13.3 | **92.9** |
| SelfCheckGPT via BERTScore | 41.3 | 26.0 | 100 | 57.1 | 40.0 | 100 | 38.7 | 24.0 | 100 | 24.6 | 14.0 | 100 |
| Zero-shot Detection | 2.5 | 100 | 1.3 | 0.0 | 0.0 | 0.0 | 0.0 | 0.0 | 0.0 | 13.3 | **100** | 7.1 |
| Reverse Validation via QG | **50.9** | **35.5** | 89.7 | 58.0 | 41.8 | 95.0 | **52.9** | **36.5** | 95.8 | 31.6 | 20.9 | 64.3 |
| Reverse Validation via EM | 50.0 | 34.7 | 89.7 | **59.0** | **42.4** | 97.5 | 44.2 | 29.6 | 87.5 | **37.7** | 25.6 | 71.4 |

Table 6: Ablation study results on the PHD benchmark. We use **Bold** to mark the best result and underline the second-best result.

*identify whether the above answer refers to {Entity}.* Nevertheless, we have observed that this approach significantly impairs the recall metric while improving the precision metric.

The aforementioned analysis demonstrates that the precision of our approach can further improve if a better automatic matching method can be available.

## 7.2 False Positive

Table 8 presents two cases of false positives, which our method mistakenly identified as factual. Due to the page limit, we only showed the non-factual part in the table.

We check the non-factual part that has been marked by the annotator to analyze why our method failed to detect the hallucination in these cases. In the case of "National Hockey League", a team joined the league in 2021, causing a change in the number of league teams. Similarly, the band Lacrimosa released a new album in 2021, changing the total number of albums. However, the training data, which serves as the knowledge source of LLMs, does not update when changes happen in the real world. Therefore, when we construct a query using outdated information, our reverse validation method may mistakenly categorize the case as factual due to successful access to the corresponding entity in the parametric knowledge base of LLMs.

This type of hallucination caused by outdated data also explains why hallucinations still occur in the *PHD-High* domain with adequate data. Unfortunately, existing zero-resource hallucination detection methods are incapable of detecting this specific type of hallucination. This limitation is inherent for zero-resource hallucination detection methods since their fundamental principle is to detect hallucinations by capturing inconsistencies in LLMs. However, when the knowledge gap in LLMs arises

| Entity | Model Answers |
|---|---|
| Maximilian II Emanuel | The entity that matches the requirements 100% is Max Emanuel, Elector of Bavaria. |
| History of Jordan | Jordan matches the requirements 100% performance |

Table 7: Two cases of False Negative. **Model Answers** means the answer we get in stage 2 of **RV via EM**. We use red to highlight the mismatched entity.

| Entity | None-Factual Part |
|---|---|
| National Hockey League | The National Hockey League (NHL) is a professional ice hockey league in North America, comprising 31 teams: 24 in the United States and 7 in Canada |
| Lacrimo | They have released 13 studio albums and are known for their dramatic and emotional live performance |

Table 8: Two cases of False Positive. We use red to highlight the non-factual information.

from outdated data rather than inadequate data, the model exhibits a high level of confidence in its responses, and almost no inconsistencies occur.

## 8 Conclusion

In this paper, we are committed to facilitating research on passage-level hallucination detection. We introduced the PHD benchmark, a new dataset for evaluating passage-level hallucination. Then, we proposed a new zero-resource hallucination detection method and demonstrated its effectiveness and robustness by comparing it with existing methods on two datasets. Finally, we manually analyzed the failure cases and revealed the shared limitation of zero-resource methods.

## Limitations

An inherent drawback of our Reverse Validation method is its inability to identify a specific type of

hallucination caused by outdated data. This is also a common flaw in current zero-resource detection methods. In addition, our approach is specifically tailored for passage-level hallucination detection and may not suitable for sentence-level detection.

## Ethics Statement

Our dataset was generated by ChatGPT and then annotated by hired workers. There are no intellectual property disputes for our data source. All annotations are collected exclusively from workers we employ, and they adhere to the relevant code of ethics. Furthermore, we pay annotators approximately $8 per hour, which is above the local minimum wage standard.

## Acknowledgements

This work was supported by National Key R&D Program of China (2021YFF0901502), National Science Foundation of China (No.62161160339), State Key Laboratory of Media Convergence Production Technology and Systems and Key Laboratory of Science, Technology and Standard in Press Industry (Key Laboratory of Intelligent Press Media Technology). We appreciate the anonymous reviewers for their helpful comments. Xiaojun Wan is the corresponding author.

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
