# OpenReview forum: "A New Benchmark and Reverse Validation Method for Passage-level Hallucination Detection"
_EMNLP/2023/Conference — EMNLP 2023 Findings_

### Official Review · Reviewer_2PhM · 2023-07-29

**Typos Grammar Style And Presentation Improvements:** 1) Line 308 - of *the* PHD benchmark …
**Soundness:** 4

**Excitement:**

3: Ambivalent: It has merits (e.g., it reports state-of-the-art results, the idea is nice), but there are key weaknesses (e.g., it describes incremental work), and it can significantly benefit from another round of revision. However, I won't object to accepting it if my co-reviewers champion it.

**Paper Topic And Main Contributions:**

The authors present two major contributions to the field of hallucination mitigation:
1) A paragraph-level dataset for zero-shot hallucination detection.
2) A method for hallucination detection named reverse validation.

Using an additional dataset, the WikiBIo-GPT3, they show their method is superior to other passage-level hallucination methods (most of them are reformed sentence-level methods).

**Questions For The Authors:**

1) You mention a very high annotator agreement in the "Quality Control" paragraph.
Did you calculate it for the whole dataset or just a small subset of fake examples? If it's the latter, can you please explain why?

**Reasons To Accept:**

1) The authors stated that they will share their code and datasets, which is much appreciated.
2) The paper is well-written and easy to follow.
3) The dataset construction is very natural, making it highly useful for truly understanding the nature of LLM's hallucinations.
4) The method is well-motivated and makes sense.
5) The results look promising, showing that both of the authors' method suggested variants are superior to their suggested baselines.

**Reasons To Reject:**

1) My main concern is the exclusive use of ChatGPT, a black-box model, in the experimental setting.
I am well aware of the authors' intention to study the behaviour of black-box models, but IMO, a community model is needed here (you can use it as a black box for keeping the main story intact) for the research community to be able to reproduce (at least some of) your results.
Recent studies showed that ChatGPT performance dramatically varies over time [1] in some tasks, which might lead to inconsistency in your main results. The first question that popped into my head when the LMvsLM baseline failed to perform is "Maybe ChatGPT is worse now than it was when they tested it?"
I am very open to discussing this issue, maybe I'm missing something? Please justify your choice.

[1] https://arxiv.org/pdf/2307.09009.pdf

2) My minor concern is that I believe your method and dataset are very specific for entity-base retrieval, but I guess you can add it to the "limitation" section.

**Reproducibility:**

2: Would be hard pressed to reproduce the results. The contribution depends on data that are simply not available outside the author's institution or consortium; not enough details are provided.

**Reviewer Confidence:**

3: Pretty sure, but there's a chance I missed something. Although I have a good feel for this area in general, I did not carefully check the paper's details, e.g., the math, experimental design, or novelty.

---

> ### Author Rebuttal · Authors · 2023-08-29
>
> Thank you for investing your time and effort in the evaluation of our work. We appreciate the opportunity to answer your concerns about our paper.
>
> **For your concern about the reproducibility of our results:** \
> We use a stable ChatGPT version in our experiment (gpt-3.5-turbo-0301 **[1]**) and set the temperature to 0, which ensures the reproducibility of our results to the greatest extent.
>
> Although we use a stable API version to ensure reproducibility, we still follow your suggestions to add experiments using the latest open-source LLM Llama-2-7b-chat-hf. Due to the time limit, we only report the result of our method, other baselines' results will be reported in the camera-ready version.
>
> | (F1/P/R) |      PHD       |    PHD_Low   |   PHD_Medium   |    PHD_High    |
> |:--------:|:--------------:|:------------:|:--------------:|:--------------:|
> |   RV-EM  |  50/34.7/89.7  | 59/42.4/97.5 | 44.2/29.6/87.5 | 37.7/25.6/71.4 |
> |   RV-QG  | 50.9/35.5/89.7 |  58/41.8/95  | 52.9/36.5/95.8 | 31.6/20.9/64.3 |
>
>
> The experiment result shows that our method is generalizable for different LLMs. We can even achieve a competitive performance using a small LLM. Codes and the hyperparameters we used in the experiments also will be released to the community to ensure the reproducibility of our results.
>
> *[1] Official document about the API version:
> https://platform.openai.com/docs/models/gpt-3-5,
> which introduces gpt-3.5-turbo-0301 as "Snapshot of gpt-3.5-turbo from March 1st 2023. Unlike gpt-3.5-turbo, this model will not receive updates, and will be deprecated on June 13th 2024 at the earliest."*
>
> **For your concern about why the LMvsLM baseline failed:** \
> We also observe the same phenomenon that ChatGPT's performance varies when we perform math reasoning tasks using different versions of ChatGPT API.
>
> However, the failure of LMvsLM is not caused by the degraded performance of ChatGPT. A version issue of ChatGPT would not result in a total inability for passage-level hallucination detection. According to our experiments and findings, the "examinee" LM tends to answer questions based on the passage content when it meets a long context (Lines 454-458). Contradictory statements will not occur in this situation. Therefore, the original LMvsLM method completely fails to detect passage-level hallucinations.
>
> We have resolved this issue successfully by restricting the "examinee" LM's access to the dialogue history (Lines 454-465). In brief, the revised version of "examinee" LLM can't see the passage content in dialogue history and only answers questions by retrieving relevant knowledge from its parameters.
>
> **For your concern about the limitation of our method:** \
> We will discuss it in the "limitation" section.
>
> Although our method detects hallucinations by constructing a prompt to retrieve the corresponding entity from LLM (Lines 325-331), it can detect any non-factual knowledge about entities, not just entities themself.  In addition, a primary use case of LLMs is answering questions and seeking factual knowledge about an entity (Lines 179-180). Therefore, the application of our method is general and broad in real-world scenarios.
>
> **For your concern about the calculation of the annotator agreement:** \
> We only calculate the annotator agreement on our fake examples (Lines 295-303) due to the high cost of human annotation. If we want to calculate it for the whole dataset, we should annotate each example with three labels, incurring a cost three times higher. It needs to be emphasized that the quality of our dataset is guaranteed as annotators are selected through rigorous criteria and qualification tests. In addition, the community can easily scale the size of the dataset at a low cost by following our annotation process.
>
> Thanks again for your time and valuable comments.

---

### Official Review · Reviewer_JYbz · 2023-07-29

**Typos Grammar Style And Presentation Improvements:** line 520
**Soundness:** 4

**Excitement:**

3: Ambivalent: It has merits (e.g., it reports state-of-the-art results, the idea is nice), but there are key weaknesses (e.g., it describes incremental work), and it can significantly benefit from another round of revision. However, I won't object to accepting it if my co-reviewers champion it.

**Paper Topic And Main Contributions:**

Current LLMs are prone to generating hallucinatory content that can be detrimental when dealing with critical tasks. This paper addresses the question of detecting such hallucinations in passage-level content generated by LLMs. Due to the lack of available resources to study this problem, the author(s) introduce the PHD dataset, a dataset of passages generated by ChatGPT that have been annotated based on the presence of hallucinatory content. To detect hallucinations in generated passages, the author(s) propose the Reverse Validation (RV) method (with two different implementations). Their best RV method can outperform baselines by at least 10% in terms of F1 on the various splits of PHD.

**Questions For The Authors:**

- If we can adapt sentence-level hallucination methods for passage-level hallucination methods, why do we need passage-level hallucination methods? Specifically, while the RV method’s numbers are convincing, it would be interesting to know the ***motivation*** for what makes the two problems distinct and worth studying separately.
- Is it possible to have an analysis of how many questions do not have key information in the RV-QG method based on a sample set? Identifying when and how often the model fails could inform future research in this direction.
- How is the “Access Database” step performed? Is it by querying the same LLM?

**Reasons To Accept:**

- This paper addresses a highly relevant problem in the current era, where language models are being considered for production-level systems to prevent the dissemination of misinformation.
- The data curation procedure employed by the authors is fairly convincing, and the reported annotator agreement scores indicate that the collected dataset is of high quality.
- The proposed RV method also outperforms baselines by a large margin on the proposed benchmark and serves as a useful baseline for future work to build upon.

**Reasons To Reject:**

Lack of transparency on collection procedure: A primary contribution of this paper is the PHD benchmark for detecting hallucinations. However, the payment structure and templates used by the author(s) have not been disclosed in this document or an ethical statement.

**Reproducibility:**

5: Could easily reproduce the results.

**Reviewer Confidence:**

4: Quite sure. I tried to check the important points carefully. It's unlikely, though conceivable, that I missed something that should affect my ratings.

---

> ### Author Rebuttal · Authors · 2023-08-29
>
> Thank you for investing your time and effort in the evaluation of our work. We appreciate the opportunity to answer your question about our paper.
>
> **Question1:Why do we need passage-level hallucination methods?**
>
> In fact, we have adapted some current leading sentence-level methods for passage-level hallucination detection in our experiment (LMvsLM and Zero-shot Detection). We have analyzed why these methods demonstrate a complete inability to detect passage-level hallucinations in Lines 450-458 and Lines 512-519 separately. Therefore, we need methods specifically designed for passage-level hallucination detection.\
> Thanks for your constructive suggestion. We want to use a paragraph to further clarify **our motivation for studying hallucination detection on the passage level**.
> ```
> Real-world applications often require passage-level hallucination detection rather than sentence-level detection. This arises from the fact that LLMs tend to furnish users with comprehensive and informative answers instead of a single sentence (Lines 56-60). Indeed, when assessing the truthfulness of a response, it is highly inefficient and costly to perform hallucination detection on each sentence within the passage [1]. In many scenarios, a judgment about the entire passage is enough, which enables a quick decision on whether to activate the retrieval module (Lines 149-155) and generate a new response. Therefore, exploring passage-level hallucination detection holds greater practical significance than exclusively concentrating on sentence-level detection. Unfortunately, the existing research mainly focuses on sentence-level detection, proposing methods and datasets that are crafted for sentence-level detection.
> ```
> **[1]: We calculate the token cost and time delay for our method and other sentence-level methods.**\
> The results are shown in the following table.
>
> |                            | Token Cost | Time Cost(second) |
> |----------------------------|------------|-----------|
> |            RV-EM           |    510.4   |    7.25  |
> |            RV-QG           |    297.3   |    2.65   |
> |       LMvsLM       |   5974.5   |   29.48   |
> | SelfCheckGPT via BertScore |    435.7   |   122.4   |
>
> Therefore, We firmly believe **it is meaningful to study passage-level hallucination detection** and we will refine our motivation in the camera-ready version.
>
> **Question2: Analysis of how many questions lost information in the RV-QG**
>
> In fact, our RV-EM variants (Lines 384-400) have resolved the question of missing key information by listing all features of the entity mentioned in response.
> We follow your suggestion to perform an analysis of how many questions lost information in the RV-QG method and how many bad cases are caused by missing key information.\
> We conducted an analysis by randomly selecting 30 samples from the output of the RV-QG method. The findings revealed that out of the 30 samples, **26 exhibited a certain degree of information loss**, while **9 samples contained incorrect labels**. Then, we analyzed **the causes of error** for these 9 examples and showed the result in the following table.
>
> | Matching Errors | Lost Key Inf. | Others |
> |:---------------:|:-------------:|:------:|
> |        3        |       4       |    2   |
>
> The analysis results clearly highlight that the primary factor influencing the performance of the RV-QG method is the loss of key information. This finding aligns with the conclusion we presented in Lines 533-535. While information loss is a common phenomenon within this method, it's important to note that **errors only emerge when substantial or essential information is missing**.
>
> **Question3: Is the “Access Database” step performed by querying the same LLM?**
>
> Yes. We use the prompt we constructed in Stage 1 to query the same LLM (Lines 335-337). Our method is a **self-check** method based on reverse validation (Line 8).\
> Nevertheless, you can also query another LLM to detect the hallucination. We follow the suggestion of reviewer 2PhM to add experiments using the latest open-source LLM **Llama-2-7b-chat-hf**. The Experimental results demonstrate that even with the utilization of a smaller LLM (**Llama-2-7b-chat-hf**) for detecting hallucinations generated by ChatGPT, our method still attains competitive performance.
>
> | (F1/P/R) |      PHD       |    PHD_Low   |   PHD_Medium   |    PHD_High    |
> |:--------:|:--------------:|:------------:|:--------------:|:--------------:|
> |   RV-EM  |  50/34.7/89.7  | 59/42.4/97.5 | 44.2/29.6/87.5 | 37.7/25.6/71.4 |
> |   RV-QG  | 50.9/35.5/89.7 |  58/41.8/95  | 52.9/36.5/95.8 | 31.6/20.9/64.3 |
>
>
> Regarding **the transparency of the collection procedure**, we will add the following ethical statement to clarify it in the camera-ready version.
> ```
> Our dataset was generated by ChatGPT and then annotated by hired workers. There are no intellectual property disputes for our data source. All annotations are collected from workers we employ and adhere to the relevant code of ethics. We pay annotators approximately $8 per hour,  which is above the local minimum wage standard
> ```
> Once again, we express our appreciation for the valuable feedback provided by you. We will add the missing section and analysis in the camera-ready version.

---

### Official Review · Reviewer_KooH · 2023-08-01

**Typos Grammar Style And Presentation Improvements:** 1. Line 324, Language -> language
**Soundness:** 3

**Excitement:**

3: Ambivalent: It has merits (e.g., it reports state-of-the-art results, the idea is nice), but there are key weaknesses (e.g., it describes incremental work), and it can significantly benefit from another round of revision. However, I won't object to accepting it if my co-reviewers champion it.

**Missing References:**

Some important work about large language models and hallucination:
1. Zhao, W. X., Zhou, K., Li, J., Tang, T., Wang, X., Hou, Y., ... & Wen, J. R. (2023). A survey of large language models. arXiv preprint arXiv:2303.18223.
2. McKenna, N., Li, T., Cheng, L., Hosseini, M. J., Johnson, M., & Steedman, M. (2023). Sources of Hallucination by Large Language Models on Inference Tasks. arXiv preprint arXiv:2305.14552.

**Paper Topic And Main Contributions:**

This paper proposes a new hallucination detection benchmark, named PHD. Correspondingly, the authors propose a self-check approach
based on reverse validation to detect factual errors automatically at the passage level.

The main contributions of this work are twofold. First, the proposed benchmark focuses on entities from Wikipedia compared to previous work. Second, the self-check hallucination detection method is performed at the passage level.

**Questions For The Authors:**

1. What is the number and domain of entities extracted from the Wikipedia dump?
2. How to conduct the qualification test and what is the labeling task? What content is shown to human labelers and what is been output?
3. What is the filtering criteria to select the final qualified workers?
4. An illustrative process of the annotation process might be helpful to understand the method.
5. In the Stage 1 of reverse validation method, how to determine and extract the entity that is questioned? In the Stage 3, the matching method is exact match or fuzzy match?
6. From the results in Table 3, the precision and recall of proposed method are much lower than other baselines (which contrasts with the result analysis) but get the best F score. Can you explain which score is more important and why does this phenomenon occur?
7. The WikiBio dataset is post-processed by aggregating sentence labels. Does it lead to an unfair comparison between the proposed method to previous methods? And is it possible to aggregate into incorrect passage label?

**Reasons To Accept:**

1. New resources. This paper presents a new hallucination detection dataset, which might be helpful for future work to evaluate LLMs. Besides, the construction process can inspire following studies in this field.
2. New method. This paper proposes a hallucination detection method at the level of passage, which is different from the previous work at the sentence level.

**Reasons To Reject:**

1. Missing details about the process of automatic data generation and human annotation.
2. Missing details about the reverse validation method.
3. Inconsistence between experimental results and analysis.

**Reproducibility:**

3: Could reproduce the results with some difficulty. The settings of parameters are underspecified or subjectively determined; the training/evaluation data are not widely available.

**Reviewer Confidence:**

5: Positive that my evaluation is correct. I read the paper very carefully and I am very familiar with related work.

---

> ### Author Rebuttal · Authors · 2023-08-29
>
> Thanks for your review and valuable questions. We appreciate the opportunity to answer your question about our paper.
>
> **For Question1:**\
> We give the version of Wikipedia dumps in the footnote of page 3, which includes 6,361,315 entities. Due to Google Search's limitations, we created a dataset of 30,000 items consisting of entities and their corresponding proxies of data volume. Our PHD Benchmark was constructed based on this dataset (Lines 224-230). We randomly sample entities to ensure the diversity of our benchmark (covering diverse domains like biology, astronomy, and geography). In addition, you can redirect to the entity's Wiki page through the link we gave in the dataset.
>
> **For Question2:**\
> We distributed questionnaires on the "Wenjuanxing" (a platform providing functions equivalent to Amazon Mechanical Turk) to conduct the qualification test (QT). The labeling task that annotators do in the QT is the same as the task they will do in the annotation stage (described in Lines 277-294), i.e. annotating the hallucination at the passage level.
>
> We defined the labels that should be outputted by human annotators in the "Labels Definition" part (Lines 244-254) and described the content and tool that annotators used in the "Two-stage Annotation" part (Lines 277-294).
> We list the input and output of human annotators in the following to clarify your confusion.
> ```
> Input: Passage generated by LLM; Wikipedia articles as references; Browser(only used in the second stage)
> Output: Label; Evidence (the content they believed to be incorrect or unverifiable)
> ```
>
> **For Question3:**
>
> We describe our criteria for annotators in the "Worker Requirements" part (Lines 255-265). We filter out unqualified annotators through their accuracy on QT to select the final qualified annotators (Lines 272-274). Questions in QT had been annotated by researchers in advance, filtering by this criteria can ensure the annotators passed QT have a high agreement with researchers
>
> **For Question4:**\
> Thanks for your suggestions. An illustrative picture of the annotation process will make it easy for the community to follow and refer to our annotation process. We will add it in the camera-ready version.
>
> **For Question5:**
>
> In Stage 1 of the reverse validation method, you can prompt the LLM to extract the entity in a zero-shot style. We give the template as follows:
> ```
> Which entity is the following passage mainly describing？ Only extract the entity name.
> Passage: {Passage}
> ```
> In practical application scenarios, we can directly extract the entity from the user's query to save token costs.
> ```
> Which entity is the following query mainly asking？ Only extract the entity name.
> Query: {query}
> ```
> In Stage 3, the matching method is an exact string match (Lines 401-403). We also discuss the drawbacks of this matching method in Section 7.1. The Precision of our approach can further improve if a better automatic matching strategy can be employed.
>
> **For Question6:**
>
> The F1 score is the harmonic mean of Precision and Recall, which can indicate the performance of the method more accurately. Even a bad method can easily attain a high score in either Precision or Recall (e.g. "All False" Baseline), but they can't achieve high scores in both simultaneously. In summary, F1 score is more important.\
> In Table 3, the two variants of our method achieve a trade-off between P and R compared to other baselines. Therefore, the proposed method gets the best F score. We firmly believe there is no inconsistency between experimental results and analysis.
>
> **For Question7:**
>
> All of the baselines are tested on the WikiBio dataset with the aggregating labels. Our strategy for aggregating labels is described below: *a passage is labeled as a hallucination if one of its sentences (or more) is a hallucination in the original dataset.*\
> There is no possibility of aggregating into incorrect passage labels if the annotation of the original dataset is correct. Therefore, our comparisons on WikiBio dataset are fair.
>
> We also want to reply to your **reasons to reject**.\
> Regarding the **missing details**, we have carefully addressed your questions and will include them, along with the missing references, in the camera-ready version. For your **third rejection reason**. We believe our answer to question 6 could resolve your concerns about the **inconsistency between experimental results and analysis**.
>
> Thanks again for your time and valuable comments.

---

### Meta-Review · Area_Chair_4SXM · 2023-09-20

**Recommendation:** 4

**Metareview:**

This paper presents a new benchmark for passage-level hallucination detection. The dataset consists of paragraphs generated by ChatGPT describing rare entities, which are annotated as being factual or not. It also presents a method for tackling the task, which essentially converts the paragraph into a question or list of entity requirements and tests whether the LM regenerates the original entity in response. Experiments show it outperforms previous approaches while running faster (the latter provided in the rebuttal).

Reviewers appreciated the dataset construction process (KooH/2PhM) and data curation (JYbz). Reviewers also appreciated the strong results of the new method. However, reviewers had some concerns about the data collection procedure regarding some missing details (KooH), lack of payment information or ethical statement (JYbz), and possible reproducibility issues with using a non-open-source model (2PhM). In the rebuttal, authors provided more information on the benchmark, point out they are using a stable ChatGPT version, and present some preliminary results with LLaMA. The proposed method seems a bit specific to the benchmark rather than being useful for detecting hallucination more broadly, but does work effectively. Overall, I think the new dataset is timely and covers an important if narrow kind of hallucination.

---

### Decision · Program_Chairs · 2023-10-07

**Decision:**

Accept-Findings

**Comment:**

This paper presents a new benchmark for passage-level hallucination detection. The dataset consists of paragraphs generated by ChatGPT describing rare entities, which are annotated as being factual or not. It also presents a method for tackling the task, which essentially converts the paragraph into a question or list of entity requirements and tests whether the LM regenerates the original entity in response. Experiments show it outperforms previous approaches while running faster (the latter provided in the rebuttal).

Reviewers appreciated the dataset construction process (KooH/2PhM) and data curation (JYbz). Reviewers also appreciated the strong results of the new method. However, reviewers had some concerns about the data collection procedure regarding some missing details (KooH), lack of payment information or ethical statement (JYbz), and possible reproducibility issues with using a non-open-source model (2PhM). In the rebuttal, authors provided more information on the benchmark, point out they are using a stable ChatGPT version, and present some preliminary results with LLaMA. The proposed method seems a bit specific to the benchmark rather than being useful for detecting hallucination more broadly, but does work effectively. Overall, I think the new dataset is timely and covers an important if narrow kind of hallucination.